# Photoactuating artificial muscle from supramolecular assembly of an overcrowded alkene-derived molecular switch

Adrien Combe [1], Shaoyu Chen [1,2] ✉, Gianni Pacella [3], Marc C. A. Stuart [1], John Y. de Boer[1], Giuseppe Portale [3] & Ben L. Feringa [1] ✉

The amplification of molecular motion along length scales for macroscopic muscle-like functions, based on supramolecular polymers, provides attractive opportunities ranging from soft actuators to responsive biomedical materials. Taking the challenge to reveal dynamic assembly parameters governing muscle functions, we present the design of a photoswitch amphiphile based on an overcrowded alkene-derived core, and developed supramolecular artificial muscles. Going from molecular motor amphiphile (**MA**) to switch amphiphile (**SA**), taking advantage of high thermal stability of the switch core, a self-recovering of bent **SA** artificial muscle is observed in post-photoactuation without external intervention. Eliminating molecular motions in **SA** artificial muscle during the post-photoactuation and aging process enables us to identify correlations between dynamic assembly transformations and macroscopic actuating functions. These findings provide insights into photoactuation and subsequent self-recovery mechanisms from the aspect of dynamic assembly process, which offers new opportunities for developing amphiphile-based supramolecular artificial muscles.

Supramolecular polymers play vital roles in living systems, ranging from key structures in cells such as actin filament, and achieving complex and responsive functions like muscles and biological motors[1–6]. The dynamic and hierarchical ordered nature of supramolecular polymers in organisms allows for the collective motion of biomolecular machines and amplification into macroscopic functions. The muscular tissue is one of the most prominent examples in which the collective and cooperative motion of actin filaments and myosin motors results in macroscopic movement[5,7]. Taking inspiration from Nature, synthetic supramolecular polymers, based on non-covalent interactions such as hydrogen bonds[8–12], electrostatic forces[13–18] and aromatic stacking[14,18–20], provide fascinating opportunities in biohybrid systems and material science[21–31]. In particular, with the cooperation of artificial molecular machines, scientists strive to explore molecular motion to develop supramolecular materials with macroscopic

mechanical movements. The key challenge of amplifying molecular motion across length scales is how to realize motion in a highly ordered or hierarchically assembled system to avoid any random energy dissipation[32]. In this context, approaches using bulk materials containing molecular machines in ordered structures, such as liquid crystals[33–36], supramolecular-covalent hybrid polymers[37,38], supramolecular or polymeric networks[39–41], and "bottom-up" hierarchically amphiphile-based supramolecular assembled hydrogels[42–45], have been demonstrated to achieve collective molecular motion, realizing the amplification of molecular motion to enable photo-driven actuators. As a "bottom-up" approach, these hierarchically amphiphile-based supramolecular systems show unique advantages, such as being self-assembled into well-organized structures due to the amphiphilic nature, control of the assembly process and functions by precise molecular design, hydrogel nature, etc. This is likely to play a

[1]Centre for System Chemistry, Stratingh Institute for Chemistry, University of Groningen, Groningen, The Netherlands. [2]School of Fashion and Textiles, The Hong Kong Polytechnic University, Hong Kong, China. [3]Macromolecular Chemistry and New Polymeric Materials, Zernike Institute for Advanced Materials, University of Groningen, Groningen, The Netherlands. ✉e-mail: shaoyu.chen@polyu.edu.hk; b.l.feringa@rug.nl

prominent role in the next-generation soft materials with sophisticated structures enabling to convert light energy into mechanical functions while providing major potential for e.g., advanced medical applications.

Our group demonstrated a soft actuator from the hierarchically supramolecular assembly of a molecular motor amphiphile (**MA**) in aqueous media, showing a muscle-like photoactuating movement (Fig. 1a)[42]. Due to the amphiphilic structure, **MA** self-assembled into worm-like micelles in aqueous media, which further assembled into a macroscopic string-shape hydrogel with anisotropic unidirectional aligned structure by drawing the **MA** solution (only 5 wt.%) into aqueous CaCl$_2$ solution using a shear-flow method. This allowed for the amplification of **MA** motion across length scales and provided an **MA** artificial muscle composed of 95% water. To develop a systematic and comprehensive system for **MA**-based artificial muscles, some modifications to the design of **MA** have been made. The hierarchical supramolecular assembled structure can be tuned by altering the length of the alkyl chains in the lower half of **MA** as well as the metal counter-cation, resulting in the control of macroscopic actuation speed[43]. Additionally, with modification of the hydrophilic end-groups from the carboxylate groups to histidine moieties, a photo- and magnetic-controlled artificial muscle was obtained[44]. Subsequently, a biocompatible **MA** artificial muscle was developed and applied as the extracellular matrix mimetic scaffold for mesenchymal stem cell culture after replacing the carboxylate groups with phosphonate groups[45]. However, all the above artificial muscles were based on the same second-generation molecular motor core. We have tried to

design and synthesize photoresponsive amphiphiles with other molecular machine cores; unfortunately, none of them can be used to develop hierarchically supramolecular artificial muscles. Therefore, extending the design of other molecular machine cores to develop supramolecular artificial muscles remains highly challenging. Furthermore, a competing reverse process due to the thermal helix inversion (THI) of **MA** may occur during the actuation, resulting in complex processes to control actuation and identify the key parameters.

To address these challenges, we anticipate that a small modification of the **MA** unit, i.e., removing the methyl group in the α-position of the central olefin in the overcrowded alkene-based core (Fig. 1a), might avoid a significant change of the molecular packing parameters in aqueous media. This would likely maintain the hierarchically supramolecular assembled features, providing a unique opportunity to develop amphiphile-based supramolecular artificial muscles. After removing the methyl group, the motor core changes to an overcrowded alkene-derived switch core (Fig. 1b). For a related switch reported by our group in the early 90s, no thermal *cis*-to-*trans* reverse isomerization was observed after the *trans*-to-*cis* isomerization upon UV irradiation[46–48], which will eliminate the effects of molecular motions on the macroscopic movements of artificial muscles in the post-photoirradiation and aging process, allowing us to identify key parameters governing the muscle-like actuating functions, e.g., the correlations between dynamic assembly transformations and macroscopic movements. Towards this goal, a thermally stable switch amphiphile (**SA**) was designed to develop supramolecular artificial

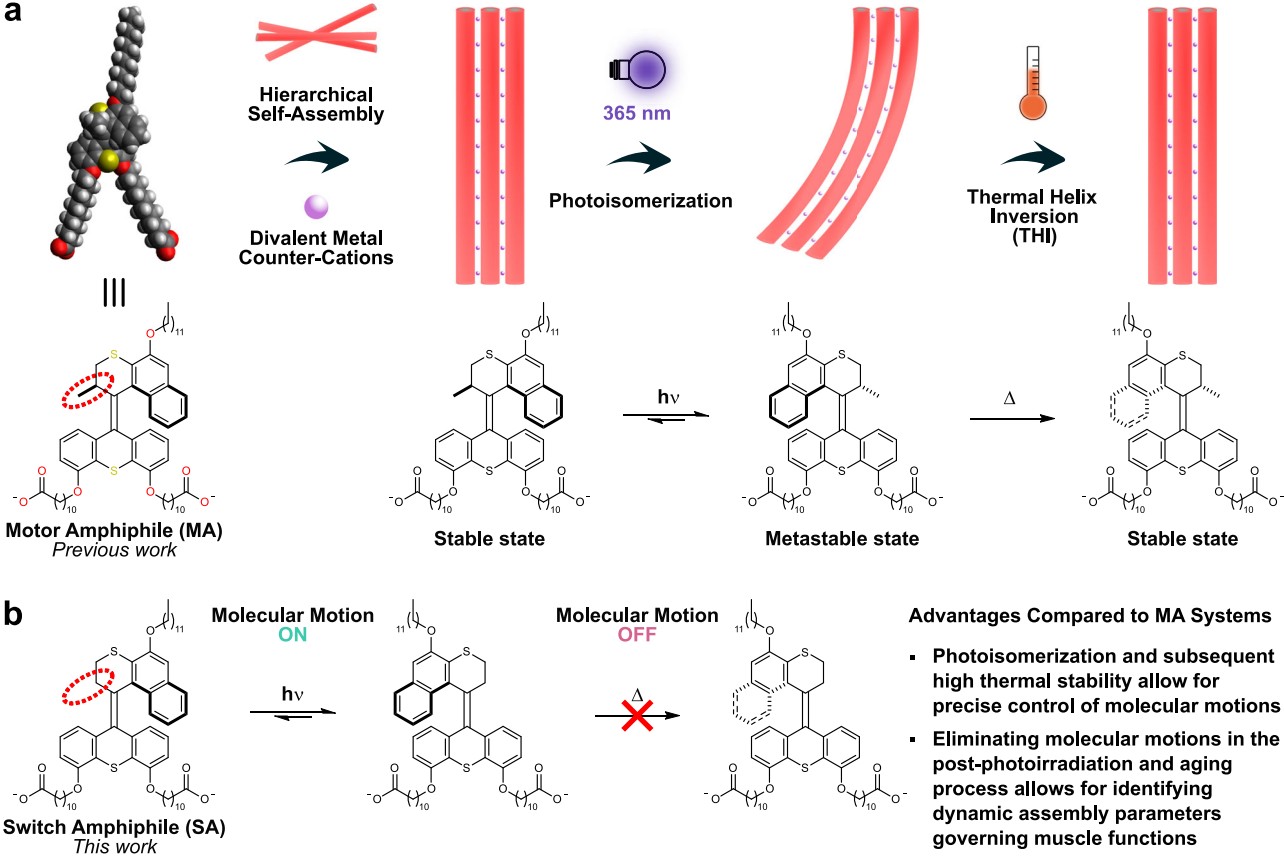

**Fig. 1 | Schematic illustration of the comparison between overcrowded alkene-derived motor (MA) and switch (SA) amphiphile-based supramolecular artificial muscles. a** Reversible actuating function of the **MA** artificial muscles induced by molecular photoisomerization and subsequent thermal helix inversion process was developed by Feringa et al. in 2017 (Copyright © 2017, The Authors)[42]. Light bulb and thermometer designed by Dr. Roza R. Weber. **b** Molecular design and switch properties of **SA**: photoisomerization and subsequent high thermal stability of **SA** allow for precise control of molecular motions, which would expect to provide new insights into the reversible actuating mechanisms of the supramolecular artificial muscles by eliminating the molecular motions during the photoactuation and aging process.

muscles (Fig. 1b). Enriching molecular liabilities for developing supramolecular artificial muscles and providing new insights into macroscopic actuation mechanisms from the aspect of dynamic assembly process represents a significant step towards developing artificial muscles with precise manipulations in structure and functions, providing major potential in future applications such as tissue engineering, i.e., stem-cell control, and soft robotics.

## Results and discussion

### Molecular synthesis

The synthetic pathway for the photoswitch **SA** is summarized in Fig. 2. The stator of **SA**, i.e., compound **7**, was synthesized according to our reported procedure[42,49]. The hydrophobic upper-half precursor was prepared starting from 2-methoxynaphthalene. Deprotonation of the starting materials with *n*-BuLi, followed by treatment with elemental sulfur, provided thiophenol **1**. Subsequently, compound **1** was converted into carboxylic acid **2** using 3-bromopropanoic acid by nucleophilic substitution. Acidic cyclization of compound **2** in methanesulfonic acid provided thiochroman-4-one **3**. Following the same synthesis pathways of deprotection and Williamson alkyl-aryl ether formation[49], the upper-half substituted with a linear dodecyl chain **5** was obtained, which was converted first in hydrazone **6** and then into diazo **9** by in situ oxidation with MnO$_2$. The Barton-Kellogg coupling reaction of diazo **9** and thioketone **8** provided the corresponding episulfide **10**. Subsequent desulfurization and hydrolysis afforded the targeted switch amphiphile (**SA**)[42]. It should be emphasized that due to the structural symmetry of the **SA** stator resulting in the lack of diastereomer formation, no direct evidence of the isomerization can be obtained by proton nuclear magnetic resonance (¹H NMR) and UV-Vis spectroscopy. In order to confirm the isomerization properties of the overcrowded alkene-based switch core (*vide infra*), a bis-methoxy switch (**BMS**, Fig. 3a) with a dissymmetric stator was designed. The detailed synthetic procedures of **SA** and **BMS**, and full characterization of new compounds, including infrared (IR) spectroscopy, ¹H and ¹³C NMR spectrometry and high-resolution mass spectrometry (HRMS), are provided in the Supplementary Information (Pages S2–S14, S22–S42, Supplementary Fig. 1–4, 20–59)[42,49,50].

### Isomerization properties of overcrowded alkene-derived switch core

Based on our previous studies on the overcrowded alkene-derived switches[46–48], (E)-**BMS** is expected to convert to (Z)-**BMS** upon UV light irradiation and no thermal reverse isomerization from (Z)-**BMS** to (E)-**BMS** occurs (Fig. 3a). To analyze the photoisomerization and thermal stability of the overcrowded alkene switch core, the

isomerization of **BMS** was examined by UV-Vis and ¹H NMR spectroscopy (Fig. 3b, c). The (E)-**BMS** and (Z)-**BMS** isomers cannot be separated by chromatography. As an alternative approach, we used the (E)-isomer-enriched solid of **BMS** for further isomerization experiments, i.e., 89% (E)-**BMS** and 11% (Z)-**BMS**, by taking advantage of the solubility difference of **BMS** in EtOH. In the UV-Vis absorption spectra of an (E)-enriched **BMS** DCM solution (Fig. 3b), a very slight increase in the absorption around 335 nm and a decrease of absorption around 305 nm are observed upon 365 nm light irradiation, showing a clear isosbestic point at 317 nm. This small change on the UV-Vis spectrum indicates a possible selective photoisomerization from (E)-**BMS** to (Z)-**BMS**. To definitely prove the photoisomerization and thermal stability of **BMS**, the interconversion of **BMS** isomers was identified by the ¹H NMR with in situ 365 nm light irradiation (Fig. 3c, d and Supplementary Fig. 5, 6). Upon 365 nm light irradiation, two distinctive proton shifts are observed in the ¹H NMR spectra of an (E)-enriched **BMS** CD$_2$Cl$_2$ solution (Fig. 3c and Supplementary Fig. 5). The proton signals of H$_{a(E)}$ and CH$_{3(E)}$ of the (E)-**BMS** shift upfield to H$_{a(Z)}$ and CH$_{3(Z)}$, demonstrating the photoisomerization from (E)-**BMS** to (Z)-**BMS**. Upon prolonging 365 nm light irradiation, photostationary state (PSS³⁶⁵) of **BMS** with an E:Z ratio of 54:46 are obtained (Fig. 3d and Supplementary Fig. 5). Notably, no transformation from (Z)-**BMS** to (E)-**BMS** is observed in the ¹H NMR spectra of the resulting CD$_2$Cl$_2$ solution kept at room temperature for 3 d (Supplementary Fig. 6), suggesting a high thermal stability of (Z)-**BMS**. The results clearly demonstrated that the *trans*-to-*cis* photoisomerization of the overcrowded alkene-derived switch core can be controlled selectively by UV light irradiation, and both isomers are thermally stable at room temperature. Compared to the UV-Vis spectra of **BMS** (Fig. 3b), similar typical absorption maxima are observed in the UV-Vis spectrum of an aqueous solution of **SA** sodium salt (Fig. 4b); therefore, 365 nm light was used to induce molecular isomerization of **SA** in aqueous media for the following experiments.

### Hierarchical supramolecular assembly

Based on the previously reported **MA** artificial muscles[42,43], the structural difference between **MA** and **SA** is the methyl group in the α-position of the central olefin. We expect that **SA** would show a similar hierarchical supramolecular assembly process in aqueous media as our **MA**-based systems[42,43]. Using 2 eq. of sodium hydroxide added to **SA** suspensions in water, followed by the subsequent annealing process (80 °C for 12 min, then cooling down overnight to room temperature), provided transparent deprotonated **SA** solutions, i.e., **SA** sodium salt solutions. Freshly prepared stock solutions of **SA** sodium salt were used to study the hierarchical supramolecular assembly of **SA** in aqueous media.

**Fig. 2 | Synthesis scheme of SA.** Schematic illustration of the synthetic pathway of photoswitch **SA**.

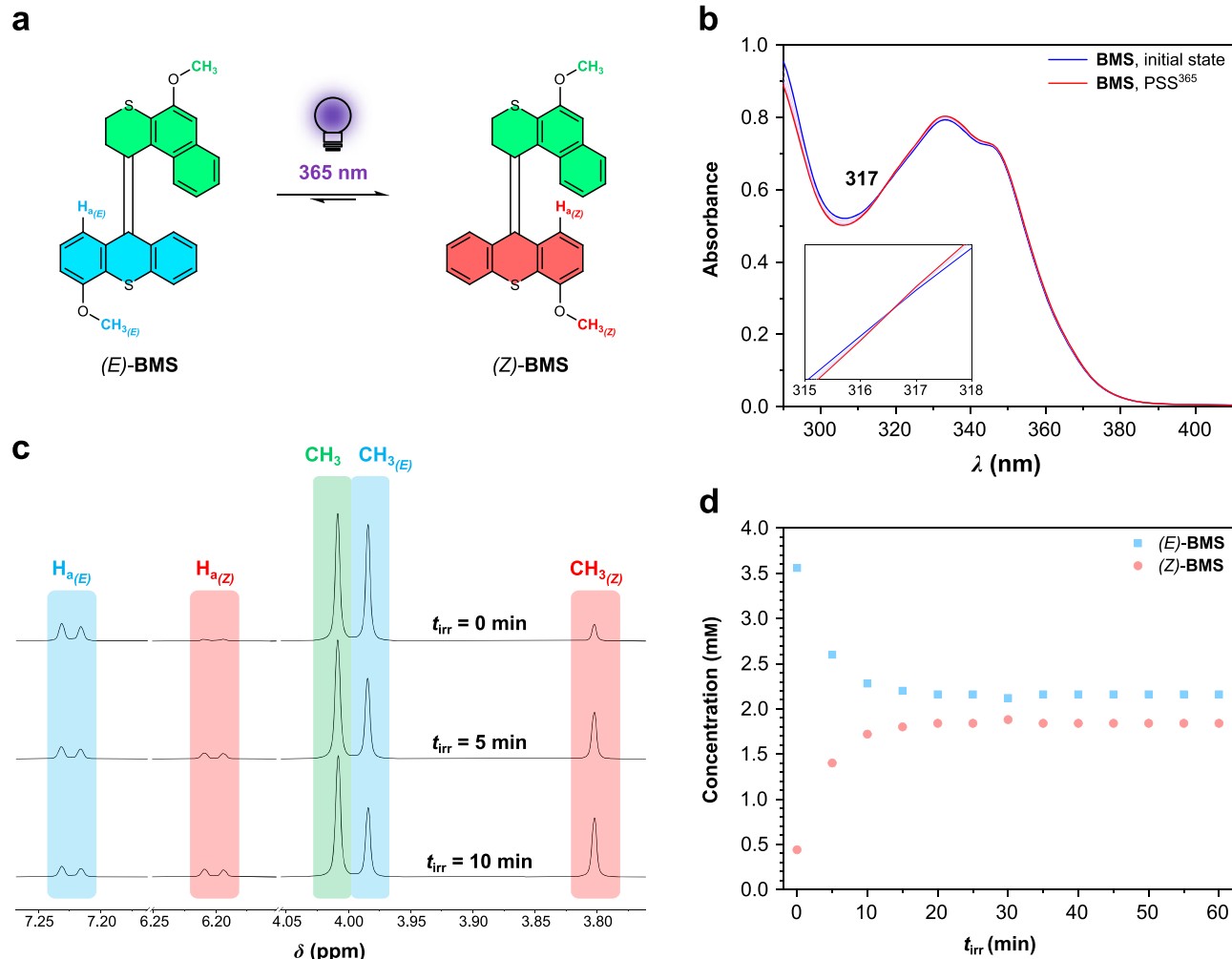

**Fig. 3 | Photoisomerization of BMS. a** Schematic illustration of the photo-isomerization reaction of **BMS**. Light bulb designed by Dr. Roza R. Weber. **b** UV-Vis spectra of an (E/Z)-**BMS** DCM solution upon 365 nm light irradiation (70 μM, 25 °C). **c** Selected regions of ¹H NMR spectral change of an (E/Z)-**BMS** solution upon

365 nm light irradiation for 10 min (500 MHz, 4.0 mM in CD₂Cl₂, 25 °C). **d** The concentration change of (E/Z)-**BMS** upon 365 nm irradiation for 1 h was determined by the integration of proton Hₐ in the ¹H NMR spectra. For the full ¹H NMR spectra, see Supplementary Fig. 5.

The critical aggregation concentration (CAC) of **SA** was determined by a Nile Red fluorescence assay (NRFA)[51,52]. When the Nile Red is encapsulated within aggregations of amphiphiles in aqueous media, a blueshift of the emission wavelength in the fluorescence spectrum can be observed, allowing it to be used as a probe for measuring the CAC. Blueshifts of the emission wavelength of Nile Red in **SA** sodium salt solutions ranging from 50 nM to 4.0 mM were measured while progressive decrease was observed when the **SA** solutions were diluted below 500 μM. This becomes significant when the **SA** concentrations are below 4.0 μM; hence, the CAC of **SA** was determined to be 1.99 μM (Supplementary Fig. 7). Given the fact of photoisomerization and high thermal stability of **SA** (Figs. 1b, 4a), UV-Vis and fluorescence spectral changes of the **SA** sodium salt solution (70 μM, above its CAC), with in situ 365 nm light irradiation and subsequent aging at room temperature, were monitored to investigate the molecular switching properties of **SA** in aggregation state. After 365 nm light irradiation for 1 h, a decrease in absorbance and slight redshift at the two shoulder peaks ($\lambda_{max}$ = 336 and 347 nm) are observed, accompanied by an obvious increase of light scattering in the UV-Vis spectra (Fig. 4b and Supplementary Fig. 8, 9), indicating a possible transformation from small-size aggregations to larger-size aggregations triggered by the photoisomerization of **SA**. From the fluorescence spectra (Fig. 4c), a significant redshift of emission peak from 412 nm to 486 nm upon

exposure to 365 nm light also indicates a possible assembly transformation[53]. To identify the assembly structure transformations of **SA** in solution upon UV-light irradiation, a **SA** sodium salt solution (5.5 mM) before and after 365 nm light irradiation was measured under cryogenic transmission electron microscopy (cryo-TEM). As shown in Fig. 4d and Supplementary Fig. 10, a mixture of small micelles and worm-like micelles in the **SA** sodium salt solution is observed, while after 365 nm light irradiation, a much higher proportion of worm-like micelles is obtained, suggesting the assembly transformations from spheroidal micelles to worm-like micelles triggered by 365 nm light irradiation (Fig. 4d and Supplementary Fig. 11). Afterward, no significant spectral change in the UV-Vis spectra and no shift of emission peak in the fluorescence spectra of the resulting 365 nm light-irradiated **SA** solution upon aging at the ambient conditions are observed (Fig. 4b, c), indicating no assembly transformations during the post-photoirradiation and aging process. Meanwhile, an increasing fluorescence intensity is detected (Fig. 4c), which may correspond to a closer molecular packing density or reassembly into worm-like micelles in the aging process. These results demonstrated that in the aggregation state, photo-driven molecular motions, i.e., helicity inversion, of **SA** (Fig. 4a) induced a disassembly process and then allowed for assembly transformations from micelles to worm-like micelles. Subsequently, since there were no continuous molecular

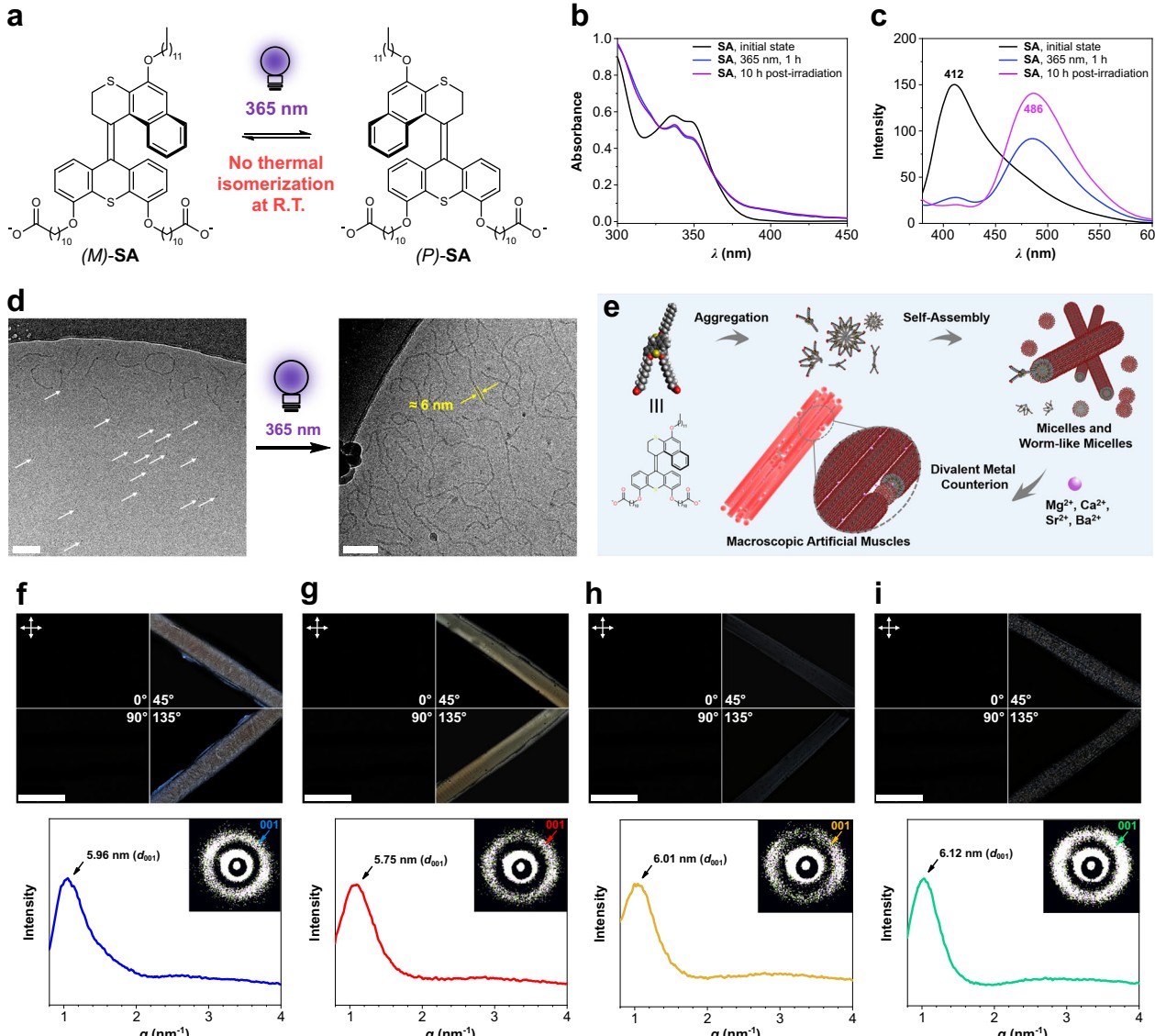

**Fig. 4 | Hierarchical supramolecular assembly of SA in aqueous media.**
**a** Schematic illustration of the photoisomerization of **SA**. Light bulb designed by Dr. Roza R. Weber. **b** UV-Vis and **c** fluorescence spectral changes of **SA** sodium salt solutions (70 μM, above CAC value) with in situ 365 nm light irradiation for 1 h and subsequent aging process at room temperature. **d** Cryo-TEM images of an **SA** sodium salt solution (5.5 mM) before and after exposure to 365 nm light for 1 h. Micelles are indicated by white arrows. Not all micelles were indicated.

Scale bar set for 100 nm. Light bulb designed by Dr. Roza R. Weber. **e** Schematic illustration of the hierarchical supramolecular assembly of **SA** across nano- to micro- and macroscopic length scales. **f–i** POM images (top) and 1D profiles along with the zoomed 2D SAXS patterns (bottom) of **SA** artificial muscles prepared from a solution of **SA** sodium salt (60 mM) in the presence of aqueous **f** MgCl$_2$, **g** CaCl$_2$, **h** SrCl$_2$, and **i** BaCl$_2$ solutions (150 mM).

motions to induce disassembly during the photoirradiation and aging process, the photo-driven out-of-equilibrium disassembled **SA** could reassemble into worm-like micelles, probably with a closer molecular packing density. Compared to our previous **MA**-based supramolecular systems[42,43,45], although some small micelles were observed in the **SA** system, the worm-like micelles were similar to **MA**'s analogs. Therefore, we expected that macroscopic noodle-like **SA** strings composed of aligned worm-like micelles could be obtained in the presence of a divalent metal counterion, allowing the development of **SA** artificial muscles with photoactuating functions (Fig. 4e).

According to our reported procedures[42], macroscopic noodle-like strings for artificial muscles were prepared by using a micropipette to manually inject the **MA** solutions into a CaCl$_2$ solution, in which a shear-flow force would induce an anisotropically orientational order of the worm-like micelles[20,42]. The electrostatic interactions between the carboxylate groups and the Ca$^{2+}$ stabilized the macroscopic artificial

muscles. Additionally, the unidirectionally aligned structure and corresponding actuating functions of **MA** can be simply modified by changing the divalent metal ions, including alkaline earth metal cations (Be$^{2+}$, Mg$^{2+}$, Ca$^{2+}$, Sr$^{2+}$ and Ba$^{2+}$)[43]. To analyze the metal counterion effects on the hierarchical supramolecular assembly in the **SA** system, the aforementioned alkaline earth metals were used as the counterions for the formation of **SA** macroscopic artificial muscles. Despite the existence of small micelles in the **SA** solution, macroscopic noodle-like **SA** artificial muscles can be obtained by extruding a 4.0 μL of **SA** sodium salt solution (60 mM) into an aqueous MCl$_2$ solution (M: Be, Mg, Ca, Sr and Ba, 150 mM) via a shear-flow method. The structural features of the **SA** artificial muscles were observed by polarized optical microscopy (POM) and further quantified by the small-angle X-ray scattering (SAXS) technique, as shown in Fig. 4f–i and Supplementary Fig. 12–17. Taking Ca$^{2+}$ as an example, the POM images of a freshly prepared **SA**-Ca$^{2+}$ artificial muscle show uniform birefringence in the

**Table 1 | Structural parameters of SA artificial muscles prepared from various metal chloride aqueous solutions (150 mm)**

| SA-M$^{2+}$ | Gray intensity | $d_{001}$ (q, nm$^{-1}$) | $d_{001}$ (d, nm) | FWHM (°) |
|---|---|---|---|---|
| **SA**-Mg$^{2+}$ | 110 | 1.05 | 5.96 | 67 |
| **SA**-Ca$^{2+}$ | 118 | 1.09 | 5.75 | 68 |
| **SA**-Sr$^{2+}$ | 33 | 1.04 | 6.01 | 63 |
| **SA**-Ba$^{2+}$ | 40 | 1.02 | 6.12 | 76 |

direction of the long axis of the muscle (Fig. 4g and Supplementary Fig. 13), indicating the formation of anisotropically orientational order of **SA** worm-like micelles. Based on the POM image, the birefringence intensity, indicated by a gray intensity value, can be obtained to evaluate the alignment degree at a micro- to millimeter-length scale, in which a stronger birefringence intensity suggests a higher macroscopic alignment degree[20,54–57]. To further quantify the structural parameters and the orientational order, i.e., the structural alignment degree, the **SA**-Ca$^{2+}$ artificial muscle, after removing the CaCl$_2$ solution and washing with double-deionized water, was placed in an aqueous media-free sealed quartz capillary for SAXS measurement. In the 1D SAXS profile (Fig. 4g), a prominent peak belonging to the diffraction from the (001) plane appears at 1.09 nm$^{-1}$, revealing the presence of a structure with characteristic spacing of 5.75 nm, which is mainly associated with the lateral packing of **SA** worm-like micelles caused by interfibrillar interactions between Ca$^{2+}$ and carboxylates[42–45]. Additionally, inspection of the 2D SAXS pattern shows that the 001 signal is not an isotropic ring. It is focused more in the direction perpendicular to the **SA** artificial muscle in the 2D SAXS image (Fig. 4e, inset), indicating that the **SA** worm-like micelles are assembled in an unidirectionally aligned order and parallel to the fiber axis. Converted from the 2D SAXS image, the azimuthal angular dependency of the intensity of the 001 diffraction signal is shown in Supplementary Fig. 17, in which the full-width half-maximum (FWHM) of the azimuthal orientation profiles can be used to quantitatively evaluate the degree of unidirectional alignment of **SA** worm-like micelles inside the volume illuminated by the X-rays. The lower the FWHM, the higher the degree of unidirectional alignment.

Following identical sample preparation and measurements, uniform birefringences along the long axis of the muscle are observed in all **SA** artificial muscles formed in the MgCl$_2$, SrCl$_2$, and BaCl$_2$ solutions (150 mm), respectively, indicating the formation of anisotropically orientational order of **SA** worm-like micelles (Fig. 4f, h, i and Supplementary Fig. 12, 14, 15). In the case of Be$^{2+}$, no stable **SA**-Be$^{2+}$ artificial muscle with anisotropic structures was obtained (Supplementary Fig. 16), which was probably attributed to the mismatch of charge and radius size of Be$^{2+}$ to the carboxylate groups in the **SA** worm-like micelles[43]. Therefore, Be$^{2+}$ was excluded in the investigations of microscopic structural features and further macroscopic actuating functions. The structural parameters of **SA** artificial muscles with various metal counterions, including the gray intensity from the POM image, as well as the maximum q-value ($q_{max}$), the interplanar spacing $d$ ($d_{001}$) and the azimuthal FWHM of the (001) plane measured by SAXS (Fig. 4f–i and Supplementary Fig. 17), are summarized in Table 1, in which the gray intensity of **SA** artificial muscle indicates the unidirectional alignment degree at a micro- to millimeter-length scale and the FWHM value reveals the alignment degree in the specific measured area at a nanometer-length scale. By considering the interplanar $d_{001}$ spacing (as derived by the Bragg's law $d_{001} = 2\pi/q_{001}$), **SA**-Ca$^{2+}$ shows the smallest spacing (5.75 nm), and the largest spacing of 6.12 nm is observed for **SA**-Ba$^{2+}$, suggesting a distinct trend following the binding constant order between low molecular weight organic carboxylates and alkaline earth metals[43], i.e., the binding constant of Ca$^{2+}$ > Mg$^{2+}$ > Sr$^{2+}$ > Ba$^{2+}$. The results of interplanar $d_{001}$ spacing indicated that a

higher binding constant between the **SA** worm-like micelles and the metal counterions resulted in a tighter supramolecular packing structure. Regarding the FWHM, comparable values are obtained in **SA**-Mg$^{2+}$, **SA**-Ca$^{2+}$, and **SA**-Sr$^{2+}$ systems, while the artificial muscle of **SA**-Ba$^{2+}$ shows a higher FWHM, which indicates a lower degree of nanoscale unidirectional alignment in the **SA**-Ba$^{2+}$ system. Furthermore, the **SA**-Ba$^{2+}$ artificial muscle also shows a low alignment degree at the millimeter-length scale based on its gray intensity value from the POM image (Table 1). Based on the POM images and SAXS measurements, a tight supramolecular packing, as in **SA**-Ca$^{2+}$ and **SA**-Mg$^{2+}$, may likely be helpful to improve the anisotropically orientational order of **SA** worm-like micelles. These results demonstrated that the structural parameters of **SA** artificial muscles can be adjusted by changing the metal counterions, which may affect their macroscopic actuating functions.

### Reversible actuating functions

To investigate the macroscopic actuating functions, an **SA** artificial muscle was prepared by a shear-flow method in a quartz cuvette containing MCl$_2$ aqueous solution (M: Mg, Ca, Sr and Ba, 150 mm), and then the bending movement for a free standing muscle from a vertical position of 0° to 90° upon 365 nm light irradiation was recorded, providing the photoactuation speed value (Fig. 5a, b and Supplementary movies 1–4). Based on the **MA** artificial muscles, the photoactuation speed can be modified by changing the metal cationic counterions due to its effects on the hierarchical supramolecular structures, in which the **MA**-Ca$^{2+}$ artificial muscle shows the highest degree of orientational alignment and the fastest photoactuation speed, indicating that a high degree of alignment accelerates the actuation speed[43]. Interestingly, with a slight modification of molecular structure, i.e., removing the methyl group in the α-position of **MA** to provide **SA**, significant differences in the metal counterion effects are observed in the **SA** artificial muscle system compared to **MA**. The fastest photoactuation speed is obtained with the **SA**-Ba$^{2+}$ artificial muscle, which shows the loosest interfibrillar packing density and the lowest degree of alignment (Fig. 5b and Table 1). As shown in Fig. 5a and Supplementary movie 4, the **SA**-Ba$^{2+}$ artificial muscle bends towards the light source from a vertical position of 0° to 90° upon 365 nm irradiation for 30 s, corresponding to an actuation speed of ≈ 3 °.s$^{-1}$. Following the identical sample preparations and measurements, the photoactuating speed order of **SA**-M$^{2+}$ artificial muscles is Ba$^{2+}$ (3.1 ± 0.2 °.s$^{-1}$) > Mg$^{2+}$ (2.5 ± 0.2 °.s$^{-1}$) > Ca$^{2+}$ (1.6 ± 0.3 °.s$^{-1}$) ≈ Sr$^{2+}$ (1.4 ± 0.1 °.s$^{-1}$), as displayed in Fig. 5b. Considering the **SA**-M$^{2+}$ supramolecular structural parameters (Table 1), it seems that the photoactuation speed of the **SA** artificial muscles is mainly determined by interfibrillar packing density, followed by the unidirectional alignment degrees of the **SA** worm-like micelles as the second factor, which is significantly different from the **MA**'s analogs where the alignment degree of the supramolecular structures was the essential parameter in determining the actuation speed[43]. This variation is probably attributed to the different properties of molecular photoisomerization. Regarding the molecular motor core, the PSS of stable-:metastable-isomer is 90:10, starting from pure stable-isomer upon 365 nm light irradiation[42,43]. However, the PSS of the switch core of the (E/Z)-**BMS** is around 54:46, starting from the (E)-enriched mixture with an E:Z ratio of 89:11 (Fig. 3). Compared to **MA**, a much lower PSS and slower photoisomerization speed is likely to be obtained in the **SA** system; therefore, a looser interfibrillar packing structure of the **SA** artificial muscles provides more free volume to enable fast molecular motions, resulting in a faster macroscopic actuation speed. Since comparable interfibrillar packing densities are observed in **SA**-Mg$^{2+}$, **SA**-Ca$^{2+}$, and **SA**-Sr$^{2+}$, their actuation speeds are determined by the second factor, i.e., the unidirectional alignment degrees at millimeter- and nanometer-length scales of the **SA** worm-like micelles. Although **SA**-Sr$^{2+}$ artificial muscle shows a slightly looser packing structure and higher nano-scale alignment degree with a smaller FWHM value compared to that of **SA**-Mg$^{2+}$, the alignment degree of **SA**-Sr$^{2+}$ at the millimeter-scale, with a gray

intensity of 33, is much lower than **SA**-Mg$^{2+}$ (gray intensity of 110), which may hinder the amplification of molecular motion at the macroscopic level, resulting in a lower photoactuation speed of **SA**-Sr$^{2+}$ (Table 1).

After 365 nm light irradiation, the bent **SA** artificial muscle returns to its initially vertical position of 0° in the cuvette by keeping it in the dark at room temperature for 48 h (Fig. 5a). Regarding the **MA** system, the bent **MA** artificial muscle returned to its initial shape only occurred by heating at 50 °C, and keeping the bent **MA** artificial muscle at room temperature for 48 h, no obvious recovering process was observed[42]. Therefore, we proposed that the metastable-**MA** transfer back to stable-**MA** via a thermal helix inversion step, resulting in the recovery shape of the artificial muscle (Fig. 1a)[42]. Given the fact that no thermal reverse isomerization at room temperature occurs in the overcrowded alkene-derived switches (Supplementary Fig. 6)[46–48], the recovery mechanism of **MA** artificial muscle is obviously not applicable to the

**SA** systems. Compared to the previous **MA** system, significant dynamic assembly properties were found in the **SA** system (Fig. 4a–e). Additionally, we observed a self-recovering process of the bent artificial muscle in post-photoirradiation and subsequent aging process that was not resulting from molecular isomerization (Fig. 5a), new reversible actuating mechanisms would be expected to be found in the **SA** system, allowing to provide a more comprehensive insight into the photoresponsive amphiphile-based supramolecular artificial muscles.

To explore the mechanisms of photoactuating and subsequent self-recovering in the aging process of **SA** artificial muscles, fluorescence spectral changes and SAXS profiles of the macroscopic **SA** artificial muscle upon in situ photoirradiation and subsequently aging at ambient conditions were monitored to analyze the assembly and structural properties in the macroscopic artificial muscle state (Fig. 5c, d and Supplementary Fig. 18, 19). Taking **SA**-Ca$^{2+}$ artificial

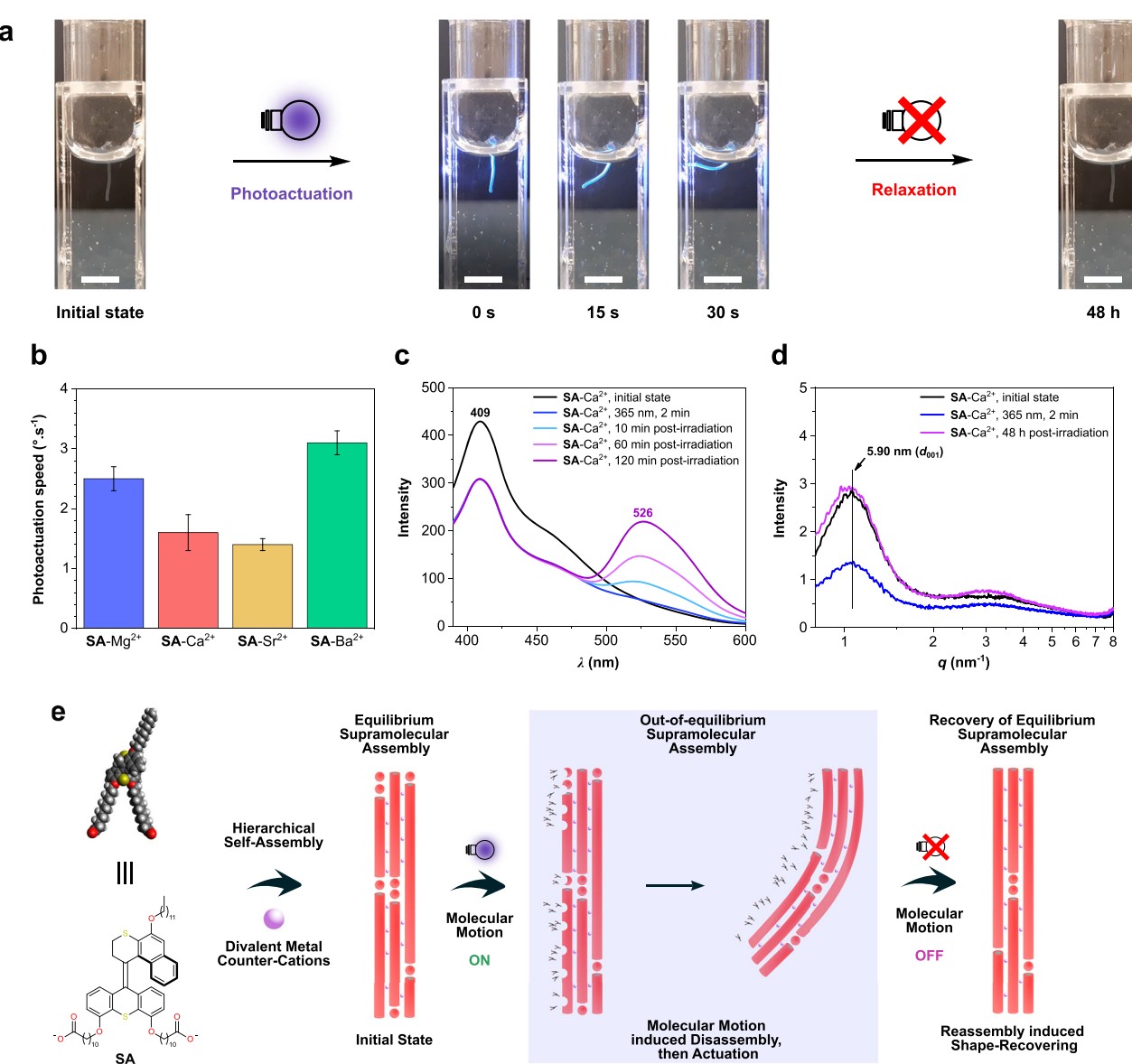

**Fig. 5 | Macroscopic artificial muscle-like actuating functions. a** Snapshots of an **SA** artificial muscle in BaCl$_2$ aqueous solution (150 mM) bending towards the light source upon 365 nm light irradiation and subsequently recovering to the initial shape after keeping in the dark for 48 h. Scale bar set for 5 mm. Light bulb designed by Dr. Roza R. Weber. **b** Comparison of photoactuation speed of **SA**-M$^{2+}$ artificial muscles prepared from various MCl$_2$ aqueous solutions (M: Mg, Ca, Sr, and Ba, 150 mM). Error bars represent triplicate experiments. **c** Fluorescence spectra and **d** 1D SAXS profiles of **SA**-Ca$^{2+}$ artificial muscle prepared from an aqueous solution of **SA** sodium salt (60 mM) in the presence of aqueous CaCl$_2$ solution (150 mM) before and after 365 nm light irradiation and subsequent aging at ambient conditions without any light exposure. **e** Proposed photoactuation and subsequent self-recovery mechanisms of **SA** artificial muscles from the aspect of the dynamic assembly process. Light bulb designed by Dr. Roza R. Weber.

muscle as an example, decreasing intensity of the emission peak at 409 nm is observed in the fluorescence spectra of **SA**-$Ca^{2+}$ artificial muscle upon exposure to 365 nm light for 2 min and during the subsequent aging process at ambient conditions, an emission peak at 526 nm becomes more and more obvious (Fig. 5c). This occurrence indicates a possible photoinduced disassembly, followed by assembly transformation and then increasing molecular packing density or reassembly in the aging process[53,58,59], which shows a consistent spectral change trend with the **SA** in solution (Fig. 4c). Further quantified changes of assembly structural parameters of the **SA**-$Ca^{2+}$ artificial muscle were detected by SAXS measurements. As shown in Fig. 5d, the intensity of the (001) plane decreases after 365 nm light irradiation for 2 min, which is also probably attributed to the partial molecular disassembly in the **SA**-$Ca^{2+}$ artificial muscle. Meanwhile, the azimuthal FWHM value of the (001) plane is detected to decrease from 117° to 106° after 365 nm light irradiation (Supplementary Fig. 19a, b), suggesting an improvement of unidirectional alignment in the **SA**-$Ca^{2+}$ artificial muscle. Given the fact that a mixture of small micelles and worm-like micelles of **SA** in solutions (Fig. 4d), the freshly prepared **SA**-$Ca^{2+}$ artificial muscle are expected to compose of small micelles and worm-like micelles. With 365 nm light irradiation, a disassembly of small micelles (Fig. 4d) results in an increasing ratio of aligned worm-like micelles in the **SA**-$Ca^{2+}$ artificial muscle, probably leading to an improvement of the average unidirectional alignment. Afterward, a recovery of the decreased peak of the (001) plane to its initial intensity before irradiation is observed after aging 48 h at ambient conditions (Fig. 5d), which also indicates a possible occurrence of reassembly, showing consistent results with the fluorescence spectra (Fig. 5c). Based on the results of fluorescence spectral changes and SAXS profiles, we propose mechanisms of photoactuating and subsequent self-recovering in the aging process of **SA** artificial muscles from the aspect of the dynamic assembly process (Fig. 5e). Specifically, a mixture of small micelles and worm-like micelles of **SA** further assembled into hierarchical supramolecular macroscopic artificial muscles in the presence of divalent metal counter ions, which were in an equilibrium assembly state. Upon 365 nm light irradiation, isomerization-induced molecular motions led to a local disassembly, further resulting in an actuating movement toward the light source[60]. Subsequently, the molecular motions can be switched off due to the high thermal stability of **SA** without UV light exposure. During the post-photoirradiation and aging process, eliminating molecular motions in the bent **SA** artificial muscle allowed the reassembly of the photo-induced disassembled **SA**, facilitating the recovery from the molecular motion-induced out-of-equilibrium assembly state to an equilibrium state. As a result, a macroscopic shape self-recovering of the bent **SA** artificial muscle was observed in the aging process. These results demonstrated that a photoactuating artificial muscle with self-recovering property in post-photoirradiation has been developed based on the hierarchical supramolecular assembly of a thermally stable photoswitch amphiphile, providing not only new insights into macroscopic actuation mechanisms from the aspect of dynamic assembly, but also a major step toward photoresponsive amphiphile-based artificial muscles with precise control over molecular motion and macroscopic actuating function.

In summary, a switch amphiphile (**SA**) based on an overcrowded alkene-derived core, in which a linear dodecyl chain attached to the upper half and two carboxylate end groups connected with linear alkyl linkers to the lower half, was designed to develop hierarchically supramolecular artificial muscles. The UV light-induced *trans*-to-*cis* isomerization and thermal stability of the switch core were confirmed by UV-Vis absorption and [1]H NMR spectroscopy. Due to its amphiphilic nature, the **SA** sodium salt self-assembled into a mixture of small spheroidal micelles and worm-like micelles at nanoscale, and further formed macroscopic micellar bundles with the electrostatic interactions between divalent metal cationic counterions and the carboxylate

groups. Detailed information on the hierarchically supramolecular assembly processes of **SA** in aqueous media was obtained by NRFA, UV-Vis and fluorescence spectroscopy, electron microscopy, and X-ray diffraction. The molecular motion between *(M)*-**SA** and *(P)*-**SA** can be amplified to endow the macroscopic **SA** string with muscle-like actuating functions, providing the development of **SA** artificial muscles. The supramolecular structural parameters and the corresponding actuation speed were modified by changing the divalent metal cationic counterions, in which **SA**-$Ba^{2+}$ artificial muscle with the loosest supramolecular packing density and the lowest degree of unidirectionally orientational structure showed the fastest photoactuation speed. In particularly, we observed dynamic assembly properties in supramolecular artificial muscle systems and a self-recovering process of the bent artificial muscle in post-photoirradiation without external intervention (i.e., not resulting from molecular isomerization). The dynamic assembly properties and no thermal reverse isomerization at room temperature of **SA** allowed us to identify new mechanisms of photoactuating and subsequent self-recovering in the aging process of **SA** artificial muscles. Notably, molecular motions from photoisomerization of **SA** induced a local disassembly in the **SA** artificial muscles, leading to photoactuating functions. Subsequently, the high thermal stability of **SA** allowed the elimination of molecular motions in the bent **SA** artificial muscle, resulting in the reassembly of disassembled **SA** and facilitating the recovery from the molecular motion-induced out-of-equilibrium assembly state to an equilibrium state. As a result, a macroscopic self-recovering of **SA** artificial muscle was observed in the aging process. The results provided new insight into macroscopic actuating mechanisms of the photoresponsive amphiphile-based supramolecular artificial muscle systems from the aspect of dynamic assembly. As a new artificial muscle based on a photoswitch amphiphile, it may open opportunities to develop next-generation soft materials with precise control over molecular motion, energy conversion, and artificial muscle-like functions.

## Methods

### Synthesis and characterizations

All commercial reagents were purchased from Aldrich, Fischer, Fluka, or Merck and were used as received unless otherwise stated. Flash chromatography was carried out using Merck silica gel 60 (230–400 mesh ASTM). Solvents for spectroscopic studies were of spectrophotometric grade (UVASOL Merck). Nuclear magnetic resonance (NMR) spectra were recorded at 25 °C on Bruker Avance 600 ([1]H: 600 MHz, [13]C: 151 MHz) and Varian Unity Plus ([1]H: 500 MHz) spectrometers. Before use, the deuterated solvents $CDCl_3$ and $CD_2Cl_2$ were treated with anhydrous $K_2CO_3$ and $MgSO_4$. Chemical shifts ($\delta$) are expressed relative to the resonances of the residual non-deuterated solvent for [1]H ($CDCl_3$: $\delta$ = 7.26 ppm, $CD_2Cl_2$: $\delta$ = 5.32 ppm, DMSO-$d_6$: $\delta$ = 2.50 ppm) and [13]C ($CDCl_3$: $\delta$ = 77.16 ppm, $CD_2Cl_2$: $\delta$ = 53.84 ppm, DMSO-$d_6$: $\delta$ = 39.52 ppm). Absolute values of the coupling constants are given in Hertz (Hz). Multiplicities are abbreviated as singlet (s), doublet (d), doublet of doublets (dd), doublet of doublets of doublets (ddd), triplet (t), doublet of triplets (dt), doublet of quadruplets (dq), multiplet (m) and broad singlet (br s). All reported [13]C NMR signals appear as single sharp peaks unless explicitly noted as broad (br). Infrared (IR) spectra were performed with a Spectrum Two FT−IR Spectrometer (PerkinElmer). Signal types are indicated as strong (s), medium (m), weak (w), broad medium (br m) and broad weak (br w). High-resolution mass spectrometry (HRMS) was done on a LTQ Orbitrap XL spectrometer with electrospray ionization (ESI) or atmospheric-pressure chemical ionization (APCI). UV-Vis measurements were performed on a HewlettPackard HP 8543 Diode Array UV-Vis Spectrophotometer in a 1 cm path length quartz cuvette. Fluorescence spectra of **SA** in solution were recorded using a JASCO FP6200 spectrofluorometer with an excitation wavelength of 365 nm in a 1 cm path length quartz cuvette. Irradiations for UV-Vis, fluorescence, and NMR kinetics experiments were carried out at 25 °C using a LED

light (Thorlabs, M365FP1, $\lambda_{max}^{em}$ = 365nm, 1.0 A, 9.8 mW). The complete synthetic procedures and characterizations of **SA** and **BMS**, including the intermediates, are detailed in the Supplementary Information.

## Standardized sample preparation of stock solution of SA in water

A few mg of switch **SA** was placed in a 0.5 mL plastic Eppendorf tube, followed by adding an aqueous solution of sodium hydroxide (4.0 M, 2.2 eq.) and double-deionized water to obtain a final **SA** concentration of 60 mM. The resulting mixture was sonicated for 5 min until it became homogeneously turbid. Subsequently, the turbid mixture was annealed at 80 °C for 12 min, and then cooled down to room temperature, without any ambient light exposure, to provide a transparent **SA** sodium salt solution.

## Nile Red fluorescence assay (NRFA)

The critical aggregation concentration (CAC) of **SA** was analyzed by incorporation of the hydrophobic solvatochromic probe Nile Red (9-diethylamino-5-benzo[$\alpha$]phenoxazinone), which shows a blueshift of the emission wavelength when it is encapsulated in hydrophobic environments. Freshly prepared Nile Red ethanol solution (250 μM) was diluted into **SA** sodium salt aqueous solutions to a final concentration of 250 nM. Each sample contains 0.1% ethanol, limiting the effect of the organic solvent on the assemblies. The fluorescence spectra of the resulting solutions containing **SA** and Nile Red were recorded at 25 °C by using a JASCO FP6200 fluorometer, which was excited with 550 nm light and measured in a wavelength range of 575–725 nm. The blueshifts were calculated by subtracting the maximum emission wavelength of Nile Red in double-deionized water from the maximum emission wavelength of the sample. Then, the CAC of **SA** was determined by plotting the obtained blueshift values against the concentrations, followed by fitting them with a *Logistic* function on a logarithmic scale of the concentrations of **SA**. The sigmoidal fitting has been performed using the software *Origin 2018*.

## Cryogenic transmission electron microscopy (Cryo-TEM)

The previously prepared solution of **SA** sodium salt (60 mM) was diluted with double-deionized water to a final concentration of 5.5 mM for the measurement. Subsequently, the identical **SA** solution was irradiated using a UV lamp (Spectroline ENB−280C 8-watt model, $\lambda_{max}^{em}$ = 365nm, 67 mA, 8 W, equipped with a LONGLIFE filter) for 1 h to investigate the effects of UV light irradiation on the assembly structures of **SA**. To observe the self-assembly structures by cryo-TEM, 2.5 μL of the sample solutions were placed on a glow-discharged holy carbon-coated grid. After blotting, the corresponding grid was rapidly frozen in liquid ethane and kept in liquid nitrogen until measurement. The grids were observed with a Gatan model 626 cryo-stage in a Tecnai T20 cryo-electron microscope operating at 200 keV. Cryo-TEM images were recorded under low-dose conditions on a slow-scan CCD camera.

## Standardized preparation of SA artificial muscle

With a micropipette and a 10 μL tip (Pipetman® Diamond tips, D10 Easy Pack, volume range: 0.1–10 μL, Gilson), 4.0 μL of **SA** sodium salt solution (60 mM) was manually drawn into an aqueous solution of $MCl_2$ (M: Mg, Ca, Sr or Ba, 150 mM) by using the shear-flow method, allowing for the formation of a noodle-like string, namely the **SA**-$M^{2+}$ artificial muscle, with a diameter of 450 ± 50 μm and an arbitrary length.

## Polarized optical microscopy (POM)

The **SA**-$M^{2+}$ artificial muscles were prepared on glass microscopic plates with a few drops of an aqueous solution of $MCl_2$ (M: Be, Mg, Ca, Sr or Ba, 150 mM). The $MCl_2$ solution was removed by adsorption with a paper tissue, and then the **SA**-$M^{2+}$ artificial muscle was washed three times with double-deionized water. The resulting **SA**-$M^{2+}$ artificial muscle was used directly in this state for the POM experiments (Nikon

model Eclipse LV100POL). The gray intensity has been measured using the software *ImageJ* [54–57].

## Fluorescence spectra of SA-$M^{2+}$ artificial muscles

Following the previously described standardized preparation, an **SA**-$M^{2+}$ artificial muscle was prepared on a JASCO EFA-382 secondary device, and its fluorescence spectral changes during UV light irradiation and subsequent aging process at room temperature were recorded using a JASCO FP6200 spectrofluorometer at 25 °C with an excited wavelength of 365 nm. The UV-light source (Thorlabs, M365L2C1, $\lambda_{max}^{em}$ = 365nm, 0.7 A, 120 mW) was placed at a distance of 26.0 ± 0.5 cm toward the **SA**-$M^{2+}$ artificial muscle for photoirradiation. During the entire measurement, the **SA**-$M^{2+}$ artificial muscle was fully immersed in the corresponding metal chloride solution.

## Small-angle X-ray scattering (SAXS)

SAXS analysis was performed at the Multipurpose Instrument for Nanostructure Analysis (MINA) beamline at the University of Groningen. The diffractometer was equipped with a Cu rotating anode ($\lambda$ = 1.5413 Å) using a sample-to-detector distance of 28.1 cm. The scattering patterns were collected using a Bruker Vantec 500 detector. The scattering angle scale was calibrated using the known position of diffraction rings from a silver behenate standard sample. The scattering intensity curves are reported as function of the modules of the scattering vector $q = 4\pi \sin(\theta)/\lambda$, with $2\theta$ being the scattering angle and $\lambda$ the wavelength of the X-rays. Following the previously described standardized preparation, an **SA**-$M^{2+}$ artificial muscle was extruded in a quartz capillary. To determine the structural parameters of the **SA**-$M^{2+}$ artificial muscle (Table 1), the material placed in the quartz capillary was washed twice with double-deionized water to remove any residual salts, and then the capillary was sealed with beeswax before measurements. To monitor the structural parameter changes upon in situ photoirradiation and the subsequent aging process (Fig. 5d), an **SA**-$Ca^{2+}$ artificial muscle was prepared and placed in a quartz capillary, partially immersed in the $CaCl_2$ aqueous solution (150 mM) to minimize drying effects, and then the capillary was sealed with beeswax before measurements. All samples were measured under vacuum.

## Photoactuation experiments

Following the previously described standardized preparation, a free-standing **SA**-$M^{2+}$ artificial muscle was prepared in a quartz cuvette (Fig. 5a). The UV light source (Thorlabs, M365L2C1, $\lambda_{max}^{em}$ = 365nm, 0.7 A, 120 mW) was placed at a distance of 26.0 ± 0.5 cm toward the artificial muscle. The light intensity at the surface of the artificial muscle was measured as 1.44 ± 0.06 mW.cm$^{-2}$. All the photoactuation experiments were performed in triplicate and recorded with a Samsung mobile phone camera. The photoactuation speed of **SA**-$M^{2+}$ was calculated from a saturated flexion angle of bending, i.e., from a vertical position of 0° to 90°, depending on a particular time. The bending angle was determined using the software *ImageJ*.

## Data availability

All data are available from the corresponding authors upon request. For additional experimental details and procedures, including spectra for all unknown isolated compounds, see supplementary information.

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

## Acknowledgements

We thank Dr. Alexander Ryabchun and Dr. Qi Zhang for their fruitful discussions and suggestions regarding the project. A.C. thanks Dr. Romain P. S. Costil, Dr. Michael P. Kathan and Dr. Wojciech Danowski for discussions and advices concerning the synthetic procedure of switch **SA**. A.C. thanks Pieter van der Meulen for his help with NMR kinetic experiments, J. L. (Renze) Sneep for all HRMS measurements, and J. (Hans) van der Velde for his assistance with the microwave reactor. A.C. thanks Dr. Roza R. Weber for drawing the light bulb and the thermometer. Financial support was granted from BIOMOLMACS, an European project funded by the European Union's Horizon 2020 research and innovation program under the Marie Skłodowska-Curie grant agreement n° 859416 to A.C., and the European Research Council (Advanced Investigator Grant n° 694345 to B.L.F.). S.C. received funding from the financial support of the China Postdoctoral Science Foundation (Grant n° 2021M69001) and PolyU Start-up Fund (Grant n° P0046278). B.L.F acknowledges the financial support from The Netherlands Organization for Scientific Research (NWO-CW), and the Ministry of Education, Culture and Science (Bonus Incentive Scheme and Gravitation Program n° 024.001.035 to B.L.F.).

## Author contributions

B.L.F. and S.C. guided the research. B.L.F. and A.C. designed the photoswitches, and together with S.C. designed the experiments. A.C. synthesized the photoswitches and performed NRFA, UV-Vis, NMR, and fluorescence experiments. A.C. and S.C. performed polarized optical microscopy and photoactuation experiments, and analyzed the data. G. Pacella has performed SAXS experiments and, together with A.C. and G. Portale, analyzed the data. M.C.A.S. has performed the cryo-TEM measurements and analyzed the data. J.Y.B. has made stock amounts of the switch **SA**, and optimized the Barton-Kellogg coupling reaction. A.C., S.C., and B.L.F. prepared the manuscript.

## Competing interests

The authors declare no competing interests.
