## [Transparent Peer Review file · Nature Communications]

Photoactuating Artificial Muscle from Supramolecular Assembly of an Overcrowded Alkene-derived Molecular Switch

Corresponding Author: Professor Ben Feringa

Version 0:

Reviewer comments:

Reviewer #1

(Remarks to the Author)

In this article, the authors report a thermal stable switch amphiphile (SA) based on an overcrowded alkene-derived core and they develop supramolecular artificial muscles using hierarchical supramolecular assembly. They also modify the actuating functions of the artificial muscle by adding different alkaline earth metal cations. I think this manuscript needs revision. I think a major issue of this paper is that it is similar to a Nature Chemistry paper published by the same group. The authors may consider the following questions and conduct necessary experiments.

1. In this work, to investigate the isomerization properties of overcrowded alkene-derived switch core (SA), the authors synthesize the model compound BMS and study its photoisomerization and thermal stability. For comparison, the authors should synthesize a model compound based on overcrowded alkene-based core (MA), which has the same structure as BMS except for a methyl group at the same position as MA. Its photochemical isomerization properties should also be studied in the same manner. This would provide a clearer demonstration of the different mechanisms of the two types of artificial muscles.
2. The authors believe that the recovery mechanism of SA artificial muscles involves a transition from the out-of-equilibrium supramolecular assembled structures to the initially stable state. The authors should provide sufficient experimental or simulation data to support this process, as it is the best crucial aspect of the article.
3. In this manuscript, the actuation speed of SA artificial muscles formed in different MCl₂ solutions is carefully studied. Additionally, artificial muscles treated with different MCl₂ solutions should exhibit varied mechanical properties. For example, under light irradiation, the maximum weight that each SA artificial muscles can lift when bent to the same angle (possibly 45°) should be determined. Based on this, the mechanical work performed by each system during this process can be calculated.
4. To demonstrate the intrinsic potential of artificial muscles, the actuation behavior of SA artificial muscles in air should be studied.
5. Can SA artificial muscles be reused? The durability of artificial muscles is crucial, so cyclic actuation testing should be included.
6. In Figure 3a, the double bond lengths of the two isomers appear to be different. The authors should standardize the bond lengths in the illustration to avoid confusing readers.
7. The structural parameters of SA artificial muscles can be adjusted by changing the metal counterions, which may probably affect their macroscopic actuating functions. Besides the metal cations used in this study, can other metal cations (maybe such as Fe³⁺) be applied in this system to modify the actuating functions?

Reviewer #2

(Remarks to the Author)

Reviewer #3

(Remarks to the Author)

In this manuscript, the authors modified a motor amphiphile, which was reported by them previously, to prepare another amphiphile that can subsequently undergo supramolecular self-assembly and reversible photoswitching.

This is an elegant work demonstrating the self-assembly of an amphiphilic molecule into artificial muscles in the macroscopic scale. With a simple change in the chemical structure by removing a methyl substituent from the overcrowded alkene-based core, the authors were able to avoid significant changes in the molecular packing, allowing the photoactuated artificial muscles to self-cover from bending. This work will be of interest to a general audience and should be suitable for publication in Nature Communications.

While the experimental work is comprehensive and convincing, this reviewer would like to understand more about the role of the metal cations in the supramolecular assembly. Fig. 1 and 4a are clear illustrations of the switch amphiphile in this work, but the organic molecules and the metal cations are not drawn to scale. In the current representations, the cations are shown to be much larger in size than the organic molecules, and might not fully reflect how the cations would affect the interfibrillar packing density. Probably the ionic radii of the cations should play a role here, and it would be better for the authors to elaborate why calcium ions can bind the most strongly, and why the smaller beryllium ions cannot give a stable and tight assembly.

Minor issue:

There is a typo on P. 10, "swith" should read "switch".

Reviewer #4

(Remarks to the Author)

This report discusses a molecular switch amphiphile that assembles into a filament-like structure, fabricated using the shear flow method in divalent metal ion solutions. This structure exhibits UV light-induced bending deformation. The switch amphiphile was synthesized by removing the methyl group from a motor amphiphile (Ref 42), resulting in a supramolecular assembly with a hierarchical structure that mimics natural muscle. Photo-bending speeds were measured to compare differences between solutions.

The introduction is well-written and the overall narrative is solid, providing clear information on the field's development. The significant efforts in molecular motor and polymer design by previous researchers are thoroughly introduced, clearly outlining the historical progress leading to this point, which I personally greatly appreciate.

The characterization and analysis are thorough and comprehensive, encompassing techniques such as NMR, UV-Vis, TEM, POM, and SAXS, making it a textbook example.

I hope these comments help improve the quality.

Main points:

1. The section on photoactuation (Fig. 5) appears somewhat sloppy compared to the previous sections. The bending experiment, in particular, is tricky. What are the absolute thicknesses of the samples in different solutions (bending is highly dependent on thickness)? What is the light intensity at the sample? What do the error bars represent—how many samples were measured, and how many times were measurements repeated? Although the authors claim 'identical sample preparation,' the samples exhibit significant variations in geometry, as shown in the accompanying videos. While bending speeds likely vary with UV intensity, the authors did not provide any data on this aspect. To clarify, these comments are not intended to request additional experiments or workload, but rather to point out that the conclusions seem premature given the simplicity of the testing.

2. The authors have emphasized several times that one of the novel aspects of their work is the observation of a self-recovery process in light-induced bending. This behavior is different from what they previously reported in the motor amphiphile system (Ref. 42). They proposed a possible mechanism for this self-recovery process, but they have not provided any supporting data.

'Then, without illumination, no thermal reverse isomerization occurs in the SA artificial muscles which allows the out-of-equilibrium supramolecular assembled structures to convert to the initially stable state, enabling the self-recovering process of the bended SA artificial muscle in post-photoirradiation.'

Given this explanation, the question arises: Why does the motor amphiphile system not exhibit the same recovery to its initial state after bending?

3. I appreciate the design of the system, from photo-switches and worm-like micelles to artificial muscle rods interlocked with counterions, which is highly hierarchical and nature-like. In general, synthetic systems for artificial muscles still fall far short of the dissipative structures found in nature, and the efficiency of human-made photoactuator lags significantly behind natural dissipative mechanisms. While different molecular systems have been developed, contributing to the growth of this important research field, I believe the discussion in this manuscript could be more comprehensive. If possible, I suggest the authors (i) compare this switch-assembled system with other existing systems that demonstrate effective photoactuation, such as azobenzene, spiropyran, and diarylethene systems; and (ii) highlight the advantages of using this actuator compared to others, for example, Ref. 42,43, which employs a similar method with a slightly different molecule.

Side.

1. The authors frequently mentioned that the self-recovery of light-induced bending occurred 'for the first time.' However, self-recovery after the light is turned off is a common behavior in light actuators, especially in systems that allow for thermal relaxation.

2. Abstract. 'better control over .. function', better compared to what?

'this precise control', what is meant by precise?

'key parameters', the only actuation characteristic provided in the manuscript is 'actuation speed.' However, many other factors are important when it comes to mimicking artificial muscles, such as modulus, stress actuation, stroke, work capacity, and repeatability.

3. Unclear to me. Shear force induces reorientation, which depends on the precursor viscosity, extrusion speed, and sodium bath conditions. But how does this lead to 'anisotropic order'? Do the assembled structures behave like rod-like liquid crystals?

4. An explanation for why there is no stable anisotropic structure formation in Be²⁺ solutions would be appreciated.

Reviewer #5

(Remarks to the Author)

Version 1:

Reviewer comments:

Reviewer #1

(Remarks to the Author)

The authors answered my questions properly. The current version can be accepted. Congratulations!

Reviewer #2

(Remarks to the Author)

I co-reviewed this manuscript with one of the reviewers who provided the listed reports. This is part of a Nature Communications initiative to promote peer review training and provide appropriate recognition to early career researchers who co-review manuscripts.

Reviewer #3

(Remarks to the Author)

The manuscript has been satisfactorily revised and my concerns have been fully addressed. I recommend acceptance of this manuscript.

made.

Point-by-point Detailed Response to the Reviewers Report and the Revisions in the Main Text Highlighted in Yellow:

Reviewer 1:

Comments:

In this article, the authors report a thermal stable switch amphiphile (SA) based on an overcrowded alkene-derived core and they develop supramolecular artificial muscles using hierarchical supramolecular assembly. They also modify the actuating functions of the artificial muscle by adding different alkaline earth metal cations. I think this manuscript needs revision. I think a major issue of this paper is that it is similar to a Nature Chemistry paper published by the same group.

Reply: The current work was indeed inspired by our previous Nature Chemistry paper mentioned by the reviewer (ref. 42), which was the first example of a soft actuator from the hierarchically supramolecular assembly of a molecular motor amphiphile in aqueous media, showing a muscle-like photoactuating movement. However, as we mentioned in the abstract and introduction (Page 2), the key challenges we solved in the current work are extending the molecular design from molecular motor amphiphile to other molecular machine cores to develop supramolecular artificial muscles. More importantly, the unique advantage of high thermal stability of the switch in this work eliminate the effects of molecular motions (from the thermal helix inversions) on artificial muscles' macroscopic movements in the post-photoactuation process. It should be noted that this unique high thermal stability of the switch allows us to identify key parameters, e.g., the establishment of correlations focusing on dynamic assembly transformations and macroscopic movements in post-photoactuation and aging process, which cannot be achieved in the molecular motor system. Following the reviewer's suggestions, we have supplemented experiments, including UV-vis and fluorescence spectroscopies, cryo-TEM, SAXS, etc., to monitor the assembly transformations in the in-situ photoirradiation and subsequent aging process (please see the responses to below Q1 and Q2 for details). The new findings in the current work provide a more comprehensive understanding of the photoactuation and subsequent self-recovery mechanism from the aspect of the dynamic assembly process in the photoresponsive amphiphile-based supramolecular artificial muscle systems.

We apologize if we were not clear enough to describe the key challenges we solved in this work. To ensure that all readers can clearly identify the key challenges and novelty of this work, we have reformulated part of the Abstract and Introduction section (highlighted in yellow in the main text) to emphasize these aspects as indicated below:

Abstract (Page 1): “Taking the challenge to reveal dynamic assembly parameters governing muscle functions, we present the design of a photoswitch amphiphile based on an overcrowded alkene-derived core and developed supramolecular artificial muscles. Going from molecular motor amphiphile (MA) to switch amphiphile (SA), taking advantage of high thermal stability of the switch core, a self-recovering of bent SA artificial muscle was observed for the first time in post-photoactuation without external intervention. Eliminating molecular motions in SA artificial muscle during the post-photoactuation and aging process enables us to identify correlations between dynamic assembly transformations and macroscopic actuating functions. These findings provide insights into photoactuation and subsequent self-recovery mechanisms from the aspect of dynamic assembly process, which offers new opportunities for developing amphiphile-based supramolecular artificial muscles.”

Introduction (Page 2): “For a related switch reported by our group in the early 90s, no thermal cis-trans reverse isomerization was observed after the trans-cis isomerization upon UV irradiation⁴⁶⁻⁴⁸, which will eliminate the effects of molecular motions on the macroscopic movements of artificial muscles in the post-photoirradiation and aging process, allowing us to identify key parameters governing the muscle functions, e.g., the correlations between dynamic assembly transformations and macroscopic actuating functions. ”..... “Enriching molecular liabilities for developing supramolecular artificial muscles and providing new insights into macroscopic actuation mechanisms from the aspect of dynamic assembly represents a significant step towards developing artificial muscles with precise manipulations in structure and functions, providing major potential in future applications such as tissue engineering, i.e., stem cell control, and soft robotics.”

Additionally, we also revised Fig. 1b to emphasize the advantages of the current switch system compared to the motor amphiphiles-based supramolecular assembled system.

The authors may consider the following questions and conduct necessary experiments:

Q1: In this work, to investigate the isomerization properties of overcrowded alkene-derived switch core (SA), the authors synthesize the model compound BMS and study its photoisomerization and thermal stability. For comparison, the authors should synthesize a model compound based on overcrowded alkene-based core (MA), which has the same structure as BMS except for a methyl group at the same position as MA. Its photochemical isomerization properties should also be studied in the same manner. This would provide a clearer demonstration of the different mechanisms of the two types of artificial muscles.

Reply: Actually, the focus of the current work is developing supramolecular artificial muscle based on a switch amphiphile with high thermal stability, which can eliminate the effects of molecular motions in supramolecular artificial muscles during the post-photoactuation and aging process, allowing us to investigate macroscopic actuation mechanisms from the aspect of dynamic assembly transformations. We don't think that synthesizing a model compound based on overcrowded alkene-based motor core (MA) to analyze its isomerization will provide useful insights for the study presented here, which is not in the scope of our main topic.

Alternatively, we believe that investigating the dynamic assembly properties of the switch amphiphile (SA) in aqueous solutions during the photoirradiation and subsequent aging process would provide more vital and straightforward insights into the SA systems. In this context, we have monitored the spectral changes of an SA sodium salt solution (70 μ M, above the CAC of SA sodium salt) upon 365 nm light irradiation and subsequent aging at ambient conditions by using UV-vis and fluorescence spectroscopy, which were added to the main text as Fig. 4b and 4c and the SI as Supplementary Fig. 6 and 7. Additionally, the assemblies of SA before and after 365 nm light irradiation were observed using cryo-TEM to confirm the solution-state morphologies transformations of SA sodium salt solution, which were also added to the main text as Fig. 4d and the SI as Supplementary Fig. 8 and 9. The results demonstrated that in the aggregation state, photo-driven molecular motions, i.e., helicity inversion, of SA induced a disassembly process and then allowed for assembly transformations from micelles to worm-like micelles. Subsequently, since there were no continuous molecular motions to induce disassembly during the aging process, the photo-driven out-of-equilibrium disassembled SA could reassemble into worm-like micelles, probably with a closer molecular packing density. The related supplemented figures and data explanations are now clearly provided in the main text (section of “Hierarchical supramolecular assembly”: page 6) and SI highlighted in yellow.

Q2: The authors believe that the recovery mechanism of SA artificial muscles involves a transition from the out-of-equilibrium supramolecular assembled structures to the initially

stable state. The authors should provide sufficient experimental or simulation data to support this process, as it is the best crucial aspect of the article.

Reply: We agree that more sufficient experimental data are needed to support the recovery mechanism of SA artificial muscles. Therefore, we have supplemented UV-vis, fluorescence spectroscopy, and cryo-TEM experiments to monitor the assembly transformations of SA in the solution state during photoirradiation and the subsequent aging process, as shown in response to Q1. Additionally, the fluorescence spectroscopy and SAXS profiles of the macroscopic SA artificial muscle upon photoirradiation and subsequently aging were added to provide more straightforward evidences, showing the dynamic supramolecular assembly transformations of SA artificial muscle, which were added in the main text as Fig. 5c and 5d and in the SI as Supplementary Fig. 16 and 17. Additionally, a schematic illustration of the proposed mechanism of the photoactuation and self-recovery of SA artificial muscle was added as Fig. 5e, hoping to provide a clearer explanation. The revised Fig. 5 is shown below:

Fig. 5 Macroscopic artificial muscle-like actuating functions. **a** Snapshots of an SA artificial muscle in BaCl₂ aqueous solution (150 mM) bending towards the light source upon 365 nm light irradiation and subsequently recovering to the initial shape after keeping in the dark for 48 h. Scale bar set for 5 mm. Light bulb designed by Roza R. Weber. **b** Comparison of photoactuation speed of SA-M²⁺ artificial muscles prepared from various MCl₂ aqueous solutions (M: Mg, Ca, Sr, and Ba, 150 mM). Error bars represent triplicate experiments. **c** Fluorescence spectra and **d** 1D SAXS profiles of SA-Ca²⁺ artificial muscle prepared from an aqueous solution of SA sodium salt (60 mM) in the presence of aqueous CaCl₂ solution (150 mM) before and after 365 nm light irradiation and subsequent aging at ambient conditions without any light exposure. **e** Proposed photoactuation and subsequent self-recovery mechanisms of SA

artificial muscles from the aspect of the dynamic assembly process. Light bulb designed by Roza R. Weber.

For more details about the data explanations, please see the revised manuscript in the section of “Reversible actuating functions” (page 11-12, highlighted in yellow).

Q3: In this manuscript, the actuation speed of SA artificial muscles formed in different MCl₂ solutions is carefully studied. Additionally, artificial muscles treated with different MCl₂ solutions should exhibit varied mechanical properties. For example, under light irradiation, the maximum weight that each SA artificial muscles can lift when bent to the same angle (possibly 45°) should be determined. Based on this, the mechanical work performed by each system during this process can be calculated.

Reply: We thank the reviewer for this suggestion. In our previous MA artificial muscles (ref. 42), we measured the lifting capacity of the MA artificial muscle when bent to ~45°. Given the fact that our amphiphile-based supramolecular artificial muscles are fragile and have weak mechanical properties, we don't think repeating the same lifting capacity experiments with the current SA artificial muscle will provide meaningful insight and novelty.

As we mentioned in the manuscript, the advantages of the SA system are able to precisely control molecular motions between on and off due to the photoisomerization and high thermal stability in the post-photoirradiation and aging process of SA, allowing us to eliminate the effects of molecular motion in post-photoactuation and identify correlations between dynamic assembly and macroscopic actuating functions. In this context, we think that it's important to keep our experiments focusing on the investigations of photoactuation and subsequent self-recovery mechanisms from the aspect of dynamic assembly, which can provide new insights into the macroscopic actuating mechanism of our amphiphile-based supramolecular artificial muscle system.

Q4: To demonstrate the intrinsic potential of artificial muscles, the actuation behavior of SA artificial muscles in air should be studied.

Reply: Since we are specifically working with hydrogel systems in the current manuscript and the major potential opportunities of our hydrogel-based artificial muscles would be biomedical applications that perform in aqueous media, such as tissue engineering, we do not feel the actuation behavior of SA artificial muscle in the air will provide new insights for actuation mechanisms of the SA artificial muscle system.

Q5: Can SA artificial muscles be reused? The durability of artificial muscles is crucial, so cyclic actuation testing should be included.

Reply: Following your suggestions, we have tested the cyclic actuation properties of the SA artificial muscles. The results showed that after aging 48 h at ambient conditions without light irradiation, the recovered SA artificial muscle lost the photoactuating functions. According to the fluorescence spectra and SAXS profiles of the macroscopic SA artificial muscle upon photoirradiation and subsequent aging process, a slightly increased molecular packing density of SA artificial muscle was observed (Fig. 5c and 5d and Supplementary Fig. 16 and 17, for more details about the data and explanations, please see the revised manuscript in the section of “Reversible actuating functions” in page 11-12 highlighted in yellow). Due to the limited free volume for SA in the recovered artificial muscle, the photoinduced molecular motions were hindered, resulting in the loss of photoactuating functions.

Q6: In Figure 3a, the double bond lengths of the two isomers appear to be different. The authors should standardize the bond lengths in the illustration to avoid confusing readers.

Reply: Following your suggestion, we have standardized the bond lengths in Fig. 3a.

Q7: The structural parameters of SA artificial muscles can be adjusted by changing the metal counterions, which may probably affect their macroscopic actuating functions. Besides the metal cations used in this study, can other metal cations (maybe such as Fe^{3+}) be applied in this system to modify the actuating functions?

Reply: Except for the metal cations we showed in the current manuscript, we also used other metal cations, including Li^+ , Na^+ , K^+ , and Sc^{3+} . However, no stable artificial muscle can be obtained from these monovalent cation solutions. Although SA artificial muscles can be obtained from the $ScCl_3$ solution, the corresponding SA- Sc^{3+} artificial muscle is in a white solid state instead of a transparent hydrogel state, which didn't show efficient photoactuation functions probably due to very limited penetration of light in the white solid-state. Considering the fact that the brown color of Fe^{3+} may hinder light penetration and the acid nature of $FeCl_3$ solution may induce the precipitate of SA sodium salt due to the protonation, we didn't try to use Fe^{3+} in this system.

Reviewer 2:

Comments:

Reply: We appreciate the reviewer's comments and suggestions.

Reviewer 3:

Comments:

In this manuscript, the authors modified a motor amphiphile, which was reported by them previously, to prepare another amphiphile that can subsequently undergo supramolecular self-assembly and reversible photoswitching.

This is an elegant work demonstrating the self-assembly of an amphiphilic molecule into artificial muscles in the macroscopic scale. With a simple change in the chemical structure by removing a methyl substituent from the overcrowded alkene-based core, the authors were able to avoid significant changes in the molecular packing, allowing the photoactuated artificial muscles to self-cover from bending. This work will be of interest to a general audience and should be suitable for publication in Nature Communications.

Reply: We are thankful for the reviewer's highly positive comments on our work and appreciate the suggestions made to improve the manuscript.

Other issues/questions:

Q1. While the experimental work is comprehensive and convincing, this reviewer would like to understand more about the role of the metal cations in the supramolecular assembly. Fig. 1 and 4a are clear illustrations of the switch amphiphile in this work, but the organic molecules and the metal cations are not drawn to scale. In the current representations, the cations are shown to be much larger in size than the organic molecules, and might not fully reflect how the cations would affect the interfibrillar packing density. Probably the ionic radii of the cations should play a role here, and it would be better for the authors to elaborate why calcium ions can bind the most strongly, and why the smaller beryllium ions cannot give a stable and tight assembly.

Reply: Following your suggestion, we have corrected the cation size in the illustrations of the previous Fig. 1 and 4a (current Fig. 4e). Indeed, the ionic radii of the counter ions are one of

the vital factors that affect the interfibrillar packing density. The binding constant between carboxylate and counter ions also plays an essential role in determining the interfibrillar packing density. It has been reported that the binding constant between low molecular weight organic carboxylates and alkali/alkaline earth metals follows the order of $\text{Ca}^{2+} > \text{Mg}^{2+} > \text{Sr}^{2+} > \text{Ba}^{2+} > \text{Li}^+ \approx \text{Na}^+ \approx \text{K}^+$ [*Thermochim. Acta* **80**, 197–208 (1984); *J. Phys. Chem. B* **108**, 4546–4557 (2004); *Coord. Chem. Rev.* **252**, 1093–1107 (2008); *J. Am. Chem. Soc.* **140**, 17724–17733 (2018)]. The smaller beryllium ions with high charge density cannot give a stable and tight assembly, and this is probably attributed to the mismatch of charge and radius size of beryllium ions to the carboxylate groups in the SA worm-like micelles [*J. Am. Chem. Soc.* **140**, 17724–17733 (2018)].

Fig. 1 Schematic illustration of the comparison between overcrowded alkene-derived motor (MA) and switch (SA) amphiphile-based supramolecular artificial muscles. **a** Reversible actuating function of the MA artificial muscles induced by molecular photoisomerization and subsequent thermal helix inversion process was developed by Feringa et al. in 2017 (Copyright © 2017, The Authors).⁴² Light bulb and thermometer designed by Roza R. Weber. **b** Molecular design and switch properties of SA: photoisomerization and subsequent high thermal stability of SA allow for precise control of molecular motions, which would expect to provide new insights into the reversible actuating mechanisms of the supramolecular artificial muscles by eliminating the molecular motions during the photoactuation and aging process.

Fig. 4 Hierarchical supramolecular assembly of SA in aqueous media. e Schematic illustration of the hierarchical supramolecular assembly of SA across nano- to micro- and macroscopic length scales.

Q2. Minor issue: There is a typo on P. 10, "swith" should read "switch".

Reply: We apologize for the typo. It has been corrected accordingly.

Reviewer 4:

Comments:

This report discusses a molecular switch amphiphile that assembles into a filament-like structure, fabricated using the shear flow method in divalent metal ion solutions. This structure exhibits UV light-induced bending deformation. The switch amphiphile was synthesized by removing the methyl group from a motor amphiphile (Ref 42), resulting in a supramolecular assembly with a hierarchical structure that mimics natural muscle. Photo-bending speeds were measured to compare differences between solutions. The introduction is well-written and the overall narrative is solid, providing clear information on the field's development. The significant efforts in molecular motor and polymer design by previous researchers are thoroughly introduced, clearly outlining the historical progress leading to this point, which I personally greatly appreciate. The characterization and analysis are thorough and comprehensive, encompassing techniques such as NMR, UV-Vis, TEM, POM, and SAXS, making it a textbook example. I hope these comments help improve the quality.

Reply: We thank the reviewer for the strong support and valuable suggestions for improving our manuscript.

Main points:

Q1. *The section on photoactuation (Fig. 5) appears somewhat sloppy compared to the previous sections. The bending experiment, in particular, is tricky. What are the absolute thicknesses of the samples in different solutions (bending is highly dependent on thickness)? What is the light intensity at the sample? What do the error bars represent—how many samples were measured, and how many times were measurements repeated? Although the authors claim 'identical sample preparation,' the samples exhibit significant variations in geometry, as shown in the accompanying videos. While bending speeds likely vary with UV intensity, the authors did not provide any data on this aspect. To clarify, these comments are not intended to request additional experiments or workload, but rather to point out that the conclusions seem premature given the simplicity of the testing.*

Reply: We apologize for the unclear descriptions of the mentioned experimental details. Actually, almost all the details were provided in the Method section (Page 12). For example, as we mentioned in the Method section about photoactuation experiments, the diameter (refers to

the thickness mentioned by the reviewer) of the artificial muscles was $450 \pm 50 \mu\text{m}$. We can only provide a diameter range instead of the absolute value because even with the same counter ion to repeat the experiments, the diameter and the geometry of corresponding SA artificial muscles slightly varied, therefore, all the experiments were repeated three times to provide an average value and error bars. Following your suggestion, we have provided a detailed description of the experimental conditions, including the light intensity, in the Method section (Page 12).

Q2. The authors have emphasized several times that one of the novel aspects of their work is the observation of a self-recovery process in light-induced bending. This behavior is different from what they previously reported in the motor amphiphile system (Ref. 42). They proposed a possible mechanism for this self-recovery process, but they have not provided any supporting data. 'Then, without illumination, no thermal reverse isomerization occurs in the SA artificial muscles which allows the out-of-equilibrium supramolecular assembled structures to convert to the initially stable state, enabling the self-recovering process of the bended SA artificial muscle in post-photoirradiation.'

Given this explanation, the question arises: Why does the motor amphiphile system not exhibit the same recovery to its initial state after bending?

Reply: To provide supporting data for the possible mechanism of photoactuation and self-recovery in post-photoactuation of the current SA artificial muscle, we have supplemented experiments, including using UV-vis, fluorescence spectroscopy, and cryo-TEM to monitor assembly transformation in solutions as well as using fluorescence spectroscopy and the SAXS technique to analyze the assembly and structural properties in the macroscopic artificial muscle state during the photoirradiation and subsequent aging process. These supplementary experimental data are now added in the main text as Fig. 4b-4c and Fig. 5c-5d and in the SI as Supplementary Fig. 6-9 and Fig. 16-17. Additionally, a schematic illustration of the proposed mechanism of the photoactuation and self-recovery of SA artificial muscle was added as Fig. 5e, hoping to provide a clearer explanation.

Regarding the MA case, after photoisomerization from stable-MA to metastable-MA, the resulting metastable-MA undergoes a thermal helix inversion process, inducing continuous molecular motions in post-photoirradiation and aging process. Therefore, after bending, the ongoing molecular motions would likely affect the assembled structures of the MA artificial muscle system and hinder the recovery to the initial equilibrium assembly state. On the contrary, after bending, the elimination of molecular motion in the SA artificial muscles in post-photoirradiation and aging process due to the high thermal stability of SA facilitates the recovery of the molecular motion-induced out-of-equilibrium assembly state, leading to a self-recovering of bended SA artificial muscle.

Q3. I appreciate the design of the system, from photo-switches and worm-like micelles to artificial muscle rods interlocked with counterions, which is highly hierarchical and nature-like. In general, synthetic systems for artificial muscles still fall far short of the dissipative structures found in nature, and the efficiency of human-made photoactuator lags significantly behind natural dissipative mechanisms. While different molecular systems have been developed, contributing to the growth of this important research field, I believe the discussion in this manuscript could be more comprehensive. If possible, I suggest the authors (i) compare this switch-assembled system with other existing systems that demonstrate effective photoactuation, such as azobenzene, spiropyran, and diarylethene systems; and (ii) highlight the advantages of using this actuator compared to others, for example, Ref. 42,43, which employs a similar method with a slightly different molecule.

Reply: (i) As we mentioned in the Introduction (Page 2), photoactuators containing other molecular machines, such as azobenzene, spiropyran, etc., were developed based on collecting molecular motions in bulk materials, including liquid crystals, polymers, etc., which were different from our hierarchically amphiphile-based supramolecular assembled system. Given the fact that the photo actuator is a very broad topic, we feel that it is important to discuss it with keeping the focus on amphiphile-based supramolecular assembled actuators in aqueous media.

(ii) Following your suggestions, we have revised some related sentences with yellow highlights in the Abstract (Page 1), the Introduction section (Page 2), and revised Fig. 1b to emphasize the advantages of the current switch system compared to the motor amphiphile-based supramolecular assembled system, namely, eliminating the effects of molecular motions on the macroscopic movements of artificial muscles in the post-photoactuation and aging process, allowing us to identify key parameters governing the muscle functions, e.g., the correlations between dynamic assembly transformations and macroscopic actuating functions, which provides new insights into macroscopic actuation mechanisms from the aspect of dynamic assembly transformations.

Side Points:

Q4. The authors frequently mentioned that the self-recovery of light-induced bending occurred 'for the first time.' However, self-recovery after the light is turned off is a common behavior in light actuators, especially in systems that allow for thermal relaxation.

Reply: Indeed, the recovery of molecular machines-based photoactuators attributed to the molecular thermal relaxation after the light is turned off is a common behavior. In our previous motor amphiphile-based supramolecular assembled artificial muscles, the thermal helix inversion of motors in the post-photoactuation process induced the recovery of the bended artificial muscles. However, it should be noted that there is no molecular thermal relaxation in the current SA artificial muscle in the post-photoactuation and aging process, as we mentioned in the manuscript, the switch core used in the current manuscript showed high thermal stability, i.e., no thermal reverse isomerization was observed after UV-light-induced isomerization. We would like to clarify that a self-recovering of photoinduced bended SA artificial muscle without any external intervention, such as thermal relaxation, was observed for the first time in the current manuscript.

Q5. Abstract. 'better control over .. function', better compared to what? 'this precise control', what is meant by precise? 'key parameters', the only actuation characteristic provided in the manuscript is 'actuation speed.' However, many other factors are important when it comes to mimicking artificial muscles, such as modulus, stress actuation, stroke, work capacity, and repeatability.

Reply: To clarify the mentioned confusion, we have revised the Abstract to provide a clear description. The revision is shown below:

“Abstract. Taking the challenge to reveal dynamic assembly parameters governing muscle functions, we present the design of a photoswitch amphiphile based on an overcrowded alkene-derived core and developed supramolecular artificial muscles. Going from molecular motor amphiphile (MA) to switch amphiphile (SA), taking advantage of high thermal stability of the switch core, a self-recovering of bent SA artificial muscle was observed for the first time in post-photoactuation without external intervention. Eliminating molecular motions in SA artificial muscle during the post-photoactuation and aging process enables us to identify correlations between dynamic assembly transformations and macroscopic actuating functions. These findings provide insights into photoactuation and subsequent self-recovery mechanisms

from the aspect of dynamic assembly process, which offers new opportunities for developing amphiphile-based supramolecular artificial muscles.”

Q6. Unclear to me. Shear force induces reorientation, which depends on the precursor viscosity, extrusion speed, and sodium bath conditions. But how does this lead to 'anisotropic order'? Do the assembled structures behave like rod-like liquid crystals?

Reply: Using shear-flow force to promote anisotropic order by aligning the worm-like micelles along the direction of the flow is a well-established and reported method. More details about this method can be found in publications, e.g., *Nat. Mater.* **9**, 594–601 (2010), *Nat. Chem.* **10**, 132–138 (2018), etc.

Q7. An explanation for why there is no stable anisotropic structure formation in Be^{2+} solutions would be appreciated.

Reply: No stable anisotropic structure formation in the Be^{2+} solution is probably attributed to the radius size mismatch of Be^{2+} to the SA worm-like micelles, which fails to provide interfibrillar electrostatic interactions in a balanced manner between Be^{2+} and carboxylate groups of SA worm-like micelles to stabilize the bundle of SA worm-like micelles [*J. Am. Chem. Soc.* **140**, 17724–17733 (2018)].

Reviewer 5:

Comments:

Reply: We thank the reviewer for the positive evaluation and detailed examination, providing valuable suggestions for improving our manuscript.